# Dynamic Task Scheduling Algorithm with Deadline Constraint in Heterogeneous Volunteer Computing Platforms

**Ling Xu [1,2,\*], Jianzhong Qiao [1], Shukuan Lin [1] and Wanting Zhang [2]**

[1] School of Computer Science and Engineering, Northeastern University, Shenyang 110819, China; qiaojianzhong@mail.neu.edu.cn (J.Q.); linshukuan@ise.neu.edu.cn (S.L.)

[2] School of Software Engineering, Dalian University of Foreign Languages, Dalian 116044, China; zhangwanting@dlufl.edu.cn

\* Correspondence: xuling@dlufl.edu.cn; Tel.: +86-158-4097-5069

**Abstract:** Volunteer computing (VC) is a distributed computing paradigm, which provides unlimited computing resources in the form of donated idle resources for many large-scale scientific computing applications. Task scheduling is one of the most challenging problems in VC. Although, dynamic scheduling problem with deadline constraint has been extensively studied in prior studies in the heterogeneous system, such as cloud computing and clusters, these algorithms can't be fully applied to VC. This is because volunteer nodes can get offline whenever they want without taking any responsibility, which is different from other distributed computing. For this situation, this paper proposes a dynamic task scheduling algorithm for heterogeneous VC with deadline constraint, called deadline preference dispatch scheduling (DPDS). The DPDS algorithm selects tasks with the nearest deadline each time and assigns them to volunteer nodes (VN), which solves the dynamic task scheduling problem with deadline constraint. To make full use of resources and maximize the number of completed tasks before the deadline constraint, on the basis of the DPDS algorithm, improved dispatch constraint scheduling (IDCS) is further proposed. To verify our algorithms, we conducted experiments, and the results show that the proposed algorithms can effectively solve the dynamic task assignment problem with deadline constraint in VC.

**Keywords:** volunteer computing; heterogeneous system; dynamic scheduling; deadline

## 1. Introduction

In recent years, volunteer computing (VC) [1] has supported diverse large-scale scientific research applications using idle resources from a large number of heterogeneous volunteer computers. VC provides not only almost free unlimited computing resources for scientific research projects, such as SETI@home [2], Folding@home [3], and ATLAS@Home [4], but also opportunities for volunteers to participate in scientific research. At the same time, increasingly more researchers have extended the unlimited computing resources provided by VC to cloud computing [5] and big data fields [6]. The network structure of VC is a master-slave distributed network computing model [7], as shown in Figure 1. The computers that provide resources are called volunteer nodes (VN), and the server is responsible for assigning tasks and collecting results.

In volunteer computing platforms, the computing power of VN is different. One of the challenges is to the algorithm of scheduling parallel tasks in such heterogeneous and dynamic platforms. At the same time, although studies have shown that assigning parallel tasks to multiple processors are NP-hard, because of its importance, many researchers have done lots of work for this problem [8–11].

However, these scheduling algorithms can't be fully applied to volunteer computing, because VN may quit at any time without any responsibility.

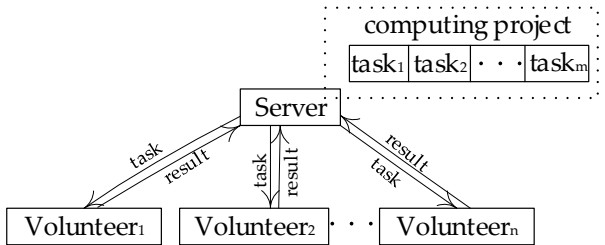

**Figure 1.** Master-slave model.

Generally speaking, volunteer computing applications are so complex that they are divided into many tasks and assigned to volunteer nodes with a hard deadline constraint [12]. If a task misses its deadline, the completion time of the whole project will be affected. For example, during the production of a chemical product, the delay of a certain ingredient will not only cause the waste of raw materials, but also postpone the delivery time. Therefore, the research on task scheduling with the deadline has great significance.

In addition, since there are not enough resources to complete all tasks in volunteer computing platforms, we mainly focus on the algorithm of completing as many tasks as possible before the deadline for each task. Similarly, Salehi et al. proposed a maximum on-time completions (MOC) [13] algorithm for task scheduling with deadline constraint for heterogeneous distributed platforms. In the MOC algorithm, a stochastic robustness measure is defined to assign tasks, and the algorithm discards tasks that miss their deadlines to maximize the number of the completed tasks. However, the MOC algorithm cannot be fully applied to the volunteer computing platform, because it does not consider suddenly offline nodes. Therefore, it is indispensable to study the dynamic task scheduling for VC platforms.

To tackle the aforementioned problems, this paper proposes two novel dynamic task scheduling algorithms with deadline constraint in heterogeneous VC platforms. To the best of our knowledge, it is the first attempt to study dynamic task scheduling with deadline constraint in the volunteer computing platform. The main contributions of this paper are summarized as follows:

(1)   A formal definition of the task assignment problem with deadline constraint in heterogeneous VC platforms for the first time.
(2)   A basic deadline preference dispatch scheduling algorithm (DPDS) that can guarantee the task with minimum deadline constraint will be computed first, a match function to select the most suitable VN in task assignment, and an improved dispatch constraint scheduling algorithm (IDCS) that selects tasks according to their priorities and utilizes a risk prediction model to improve the execution efficiency of the application.
(3)   A comprehensive evaluation of the proposed algorithms and a comparison with existing algorithms.

The remainder of this paper is organized as follows: The next section presents the related work. Section 3 introduces the definition of the problems. Section 4 illustrates our task scheduling algorithms. Section 5 gives the experimental results and analysis of the proposed task scheduling algorithms. Section 6 concludes this paper.

## 2. Related Work

In this section, we summarize the related work of task scheduling algorithm. Firstly, we introduce task scheduling algorithms in other distributed computing systems. Secondly, we introduce task assignment algorithms in volunteer computing platforms.

### 2.1. Task Scheduling Algorithms in Other Distributed Computing Systems

Because effective task allocation algorithm can improve the performance of distributed systems, many researchers have done a lot of works in this area. These task scheduling algorithms can be roughly divided into two categories, static scheduling algorithms and dynamic scheduling algorithms.

Generally speaking, static algorithms use directed acyclic graphs (DAG) [14] to represent task priorities in task scheduling. Topcuoglu et al. [15] proposed two static task scheduling algorithms for heterogeneous distributed computing, which are called earliest-finish-time (HEFT) algorithm and the critical-path-on-a-processor (CPOP) algorithm. The HEFT algorithm chooses tasks with the highest priority at each step to allocate tasks and the CPOP algorithm calculates task priority according to DAG. The ultimate goal of the HEFT is to minimize the completion time of all tasks. The Min-min algorithm [16] extends the HEFT algorithm and uses sophisticated heuristics at each level to reduce the probability of catching local minima. Poola et al. [17] proposed a robust task scheduling algorithm based on deadline and budget constraints for cloud computing, which can minimize the total completion time and the cost under deadline constraint. However, the key issue of these prior studies is that they didn't consider the dynamic of distributed computing. Because of the dynamic characteristics of volunteer computing platforms, the static scheduling algorithms can't be fully applied to such platforms, and cannot make full use of the computing resources of the system.

To improve the utilization rate of system resources, researchers have proposed many dynamic scheduling algorithms. Zomaya et al. [18] proposed a dynamic load-balancing algorithm to improve the utilization of computing resources; the experimental results show that their algorithm can achieve a near-optimal task allocation. On this basis, Page et al. [19] proposed an improved dynamic scheduling algorithm which operates in a batch fashion and uses a genetic algorithm to minimize the total completion time. In addition, the greedy algorithm [20] is used to task allocation. However, these algorithms did not take the deadline into account, so it is necessary to design a heterogeneous dynamic scheduling algorithm with the deadline.

The MOC algorithm [13] was proposed to solve the dynamic scheduling problem with deadline constraint in heterogeneous distributed computing, as mentioned before. However, the objective of MOC is that each task must be completed before its deadline, which is different from our work: To complete as many tasks as possible within their respective deadline constraints. Moreover, there are some dynamic task assignment methods with deadline constraints for special application scenarios [21–23]. Since application scenarios differ from volunteer computing, these algorithms can't be applied to volunteer computing.

### 2.2. Task Scheduling Algorithms in Volunteer Computing Platforms

Anderson et al. [24,25] introduced the dynamic of volunteer computing systems, and they proposed a basic task scheduling algorithm according to the attributes of the tasks and computing resources, such as deadline of the tasks, computing power of volunteer nodes, and the number of tasks to arrive, etc. The system constraints proposed in their system are the same as our work. In contrast to their work, the objective of our work is to maximize the number of completed tasks.

To improve the performance of volunteer computing, it is necessary to design an appropriate task scheduling algorithm to complete as many tasks as possible. This type of algorithms is called throughput driven task scheduling algorithm, which has been extensively studied in previous work. For example, Guler et al. [26] proposed a task allocation algorithm to maximize the number of task completions under monetary budget constraint. They also verify the effectiveness of their algorithm under the price of the electricity consumed by their peers. To make full use of computing resources and increase the percentage of workflows that meet the deadline, Ghafarian et al. [27,28] proposed a workflow scheduling algorithm. The proposed workflow scheduling algorithm partitions a workflow into sub-workflows to minimize data dependencies among the sub-workflows. At the same time, the experimental results show that the proposed algorithm increases the percentage of workflow that meets the deadline with a factor of 75%. Sebastio et al. [29] proposed a framework to allocate tasks

according to different policies in volunteer cloud systems. To maximize the number of the completed tasks, they take into account several different aspects. Then they provided a distributed optimization approach relying on the alternating direction method of Multipliers algorithm (ADMM) and the results show that the ADMM algorithm has a good performance in a real environment.

By comparing the above studies, we conclude that algorithms mainly focus on maximizing the number of task completions under different constraints. In contrast to prior works, this paper focuses on dynamic task scheduling which can react quickly in some situations, such as suddenly offline, new arrivers etc. At the same time, this paper intends to maximize the number of task completions considering the deadline for each task.

## 3. Problem Description

This paper studies the dynamic task allocation method with deadline constraint in the volunteer computing platforms. The notations used in the paper are summarized in Table 1.

**Table 1.** Summary of notations.

| Notation | Notation Meaning |
|---|---|
| $t_1, t_2$ | Task |
| $T_l$ | The set of tasks at time $l$ |
| $n_1, n_2$ | Volunteer node |
| $N_l$ | The set of volunteer nodes at time $l$ |
| $t_i.cost$ | Accumulative execution time for task $t_i$ |
| $t_i.deadline$ | Deadline constraint for task $t_i$ |

Given a number of tasks in VC platforms, denoted by the set $T = \{t_1, t_2, \ldots, t_m\}$. Suppose that each task $t_i$ can be completed by any node $n_j$ or several other nodes in the VC platforms. For easy expression, two unified concepts are first introduced in the VC platforms. The server hour means the unit time of the server. One unit means the number of tasks completed within one server hour.

The concepts of task and node are given as follows:

**Definition 1 (task).** *Task $t_i$ is a double dimension array that is denoted by ($t_i.cost$,$t_i.deadline$). The unit of $t_i.cost$ is called server hour, meaning that $t_i$ can be finished on the server for $t_i.cost$ hours; $t_i.deadline$ means the constraint of task $t_i$, whose unit is hours. If the task $t_i$ begins at time $l_1$, the deadline constraint of task $t_i$ means that task $t_i$ must be completed before $l_1 + t_i.deadline$.*

For example, as shown in Figure 2a, $t_3.cost = 5$, which means that it will take five server hours to complete the task $t_3$. If the deadline constraint of task $t_3$ is three hours and begins at time $l_1$, it means that task $t_3$ must be finished before $l_1 + 3$.

**Definition 2 (node).** *Given a volunteer node set, denoted by the set $N = \{n_1, n, \ldots, n_j\}$, the computing power of each volunteer node $n_j$ is denoted by $n_j.ablitlity$, it means that the number of tasks the node $n_j$ can complete in an hour is $n_j$. ability units.*

For example, as shown in Figure 2b, $n_1$. *ability* = 1.3, which means that the volunteer node $n_1$ can complete the number of tasks in an hour is 1.3 times unit.

A dynamic task scheduling with deadline constraint at $l_1$ moment can be denoted by $t_i$. *assign* = $\{(n_1, t_1, l_1), (n_2, t_1, l_1), \ldots (n_s, t_1, l_1)\}$.

List of tasks at time $l_1$ is shown in Figure 2a, List of volunteer nodes at time $l_1$ is shown in Figure 2b and the task allocation process is shown in Figure 2c. Task $t_1$ is taken as an example: The server starts to allocate computing resource to the task $t_1$ at the initial time $l_1$ and the task $t_1$'s allocation is denoted by $t_1.assign = \{(n_1, t_1, l_1), (n_2, t_1, l_1), (n_3, t_1, l_1), (n_4, t_1, l_1), (n_5, t_1, l_1), (n_6, t_1, l_1), (n_1, t_1, l_1 + 1), (n_2, t_1, l_1 + 1)\}$.

The calculation cost of $t_1$ is seven units and $n_1$ is scheduled to complete two-hour task $t_1$ by the server. It is known that the number of tasks $n_1$ can complete in an hour is 1.3 times unit. So, $n_1$ can complete 2.6 units in two hours. By analogy, the total computing resources allocated to $t_1$ can complete a total of eight units, which are more than seven units of $t_1$. Furthermore, because the deadline constraint of $t_1$ is two hours, so, $t_1$ can be completed. Similarly, the computing resources allocated to $t_2$ is $t_2.assign = \{(n_3, t_2, l_1 + 1), (n_4, t_2, l_1 + 1), (n_5, t_2, l_1 + 1), (n_6, t_2, l_1 + 1), (n_1, t_2, l_1 + 2)\}$. The calculation cost of $t_2$ is four units, and the deadline constraint of $t_2$ is three hours, so $t_2$ can be completed. The computing resources allocated to $t_3$ is $t_3.assign = \{(n_2, t_3, l_1 + 2), (n_3, t_3, l_1 + 2), (n_4, t_3, l_1 + 2), (n_5, t_3, l_1 + 2), (n_6, t_3, l_1 + 2), (n_1, t_3, l_1 + 3)\}$. The calculation cost of $t_3$ is five units and the deadline constraint of $t_3$ is three hours. The deadline constraint of $t_3$ cannot be satisfied, therefore $t_3$ cannot be completed. By analogy, it can be concluded that $t_4$ can be completed and $t_5$ cannot be completed. Finally, according to the task allocation of Figure 2c, the tasks that can be completed before deadline constraint are $t_1$, $t_2$ and $t_4$.

| $T$ | $t_i.cost$ | $t_i.deadline$ |
|---|---|---|
| $t_1$ | 7 | 2 |
| $t_2$ | 4 | 3 |
| $t_3$ | 5 | 3 |
| $t_4$ | 3 | 4 |
| $t_5$ | 3 | 4 |

(a)

| $N$ | $n_j.ability$ |
|---|---|
| $n_1$ | 1.3 |
| $n_2$ | 1.2 |
| $n_3$ | 0.5 |
| $n_4$ | 1 |
| $n_5$ | 0.7 |
| $n_6$ | 0.8 |

(b)

(c)

**Figure 2.** An example of task scheduling at time $l_1$. (**a**) List of tasks at time $l_1$; (**b**) list of volunteer nodes at time $l_1$; (**c**) task allocation process at time $l_1$.

In a dynamic network environment, the volunteer nodes are updated hourly. List of tasks at time $l_1 + 1$ is shown in Figure 3a, and the list of volunteer nodes at time $l_1 + 1$ is shown in Figure 3b. At time $l_1 + 1$, the node $n_2$ goes offline, the node $n_7$ goes online. Obviously, the task assignment process in Figure 2c cannot meet the requirements of dynamics, so it is necessary to design a corresponding dynamic task assignment algorithm. The task scheduling algorithm of this paper will be described in the following section.

In VC platforms, the set of volunteer nodes and task sets are updated dynamically over time. In this paper, the objective of the task allocation is to solve the dynamic task scheduling problem and maximize the number of completed tasks before deadline constraint.

| T | $t_i.cost$ | $t_i.deadline$ |
|---|---|---|
| $t_1$ | 1.5 | 1 |
| $t_2$ | 4 | 2 |
| $t_3$ | 5 | 2 |
| $t_4$ | 3 | 3 |
| $t_5$ | 3 | 3 |
| $t_6$ | 2.5 | 1 |

| N | $n_j.ability$ |
|---|---|
| $n_1$ | 1.3 |
| $n_2$ | 1.2 |
| $n_3$ | 0.5 |
| $n_4$ | 1 |
| $n_5$ | 0.7 |
| $n_6$ | 0.8 |
| $n_7$ | 1 |

(a)  (b)

**Figure 3.** Task list and volunteer nodes list at time $l_1 + 1$. (**a**) List of tasks at time $l_1 + 1$; (**b**) list of volunteer nodes at time $l_1 + 1$.

## 4. Algorithm Description

In this section, we introduce the DPDS algorithm and the IDCS algorithm in detail.

### 4.1. The Deadline Preference Dispatch Scheduling (DPDS) Algorithm

The DPDS algorithm is a dynamic task scheduling algorithm based on deadline constraint priority. Firstly, the DPDS algorithm sorts the tasks in ascending order according to their deadline and sorts the VN in descending order according to their computing power. Secondly, the DPDS algorithm adopts a matching function to select the most suitable volunteer node to assign a task, which make the computing resource will be freed in the least possible amount of time. Consequently, the DPDS algorithm can ensure that the task with nearest deadline constraint is completed first, and free computing resource in the least possible amount of time. The DPDS algorithm is described in detail in Algorithm 1.

Significantly, in the DPDS algorithm, we use a monitoring mechanism, which is triggered to meet the following two conditions simultaneously.

- The volunteer computing platform has been running for an hour.
- There are new nodes or tasks arrived at the VC platform.

---

**Algorithm 1. The DPDS Algorithm Taskassign($T_l,N_l,l$)**

---

**Input:** task set $T_l$, volunteer nod set $N_l$, the current time $l$
**Output:** final task assignment set $T.assign$.
1.     Wait until the monitoring mechanism is trigged
2.     $T.assign = \emptyset$
3.     Sort $T_l$ in ascending order according to the task's deadline at time $l$
4.     Sort $N_l$ in descending order according to the node's computing power at time $l$
5.     **While** $T_l \neq \emptyset$ **do**
6.        Take the first task from $T_l$ to $t'$
7.        Call the **match($t'$, $N_l$, $l$)** // select the most suitable volunteer node to $t'$
8.        Delete $t'$ from $T_l$
9.        $T.assign = T.assign + t'.assign$
10.    **End While**
11.    **Return** $T.assign$

---

If the monitoring mechanism is triggered, to make full use of the computing resources provided by the idle VN, we adopt the match function to select the suitable VN that wastes the least amount of computing resources in the DPDS algorithm. The match function is described in detail in Algorithm 2.

For example, the list of tasks at time $l_1$ is shown in Figure 2a and list of volunteer nodes at time $l_1$ is shown in Figure 2b. According to the first to fourth lines of Algorithm 1, at time $l_1$, $T_1 = \{t_1, t_2, t_3, t_4,$

$t_5$}, $N_l$ = {$n_1$, $n_2$, $n_4$, $n_6$, $n_5$, $n_3$}. Take $t_1$ as an example, according to the sixth line of Algorithm 1, it can be seen that $t_1$ is allocated first. According to Algorithm 2, at time $l_1$, $t_1.assign$ = {($n_1$, $t_1$, $l_1$), ($n_2$, $t_1$, $l_1$), ($n_3$, $t_1$, $l_1$), ($n_4$, $t_1$, $l_1$), ($n_5$, $t_1$, $l_1$), ($n_6$, $t_1$, $l_1$)}. As can be seen from the task assignment above, there is still 1.5 units left in task $t_1$, which needs to be allocated at time $l_1$ + 1. At time $l_1$ + 1, both the task list and the node list are updated, which are shown in Figures 3a and 3b, so the monitoring mechanism is triggered. According to the first to fourth lines of Algorithm 1, at time $l_1$ + 1, $T_1$ = {$t_1$, $t_6$, $t_2$, $t_3$, $t_4$, $t_5$}, $N_l$ = {$n_1$, $n_4$, $n_7$, $n_6$, $n_5$, $n_3$}. According to the sixth line of Algorithm 1, it can be seen that $t_1$ is allocated first. According to Algorithm 2, at time $l_1$ + 1, $t_1.assign$ = {($n_1$, $t_1$, $l_1$ + 1), ($n_3$, $t_1$, $l_1$ + 1) }. By analogy, we assume that tasks and nodes are updated only at time $l_1$ + 1, so the rest of the task assignments are shown in Figure 4. Figure 4 shows $t_3$ and $t_5$ cannot be completed within the deadline.

---

**Algorithm 2. The Match($t'$, $N_l$, $l$)**

---

**Input:** the task $t'$, volunteer node set $N_l$, the current time $l$
**Output:** task assignment set $t'.assign$

1.      Initialize $t'.assign$ = $\emptyset$; $t'.remain$ = $t'.cost$; $t'.hasassign$ = 0
2.      $total\_ability$ = the total computing power of the set $N_l$ at time $l$
3.      **While** $t'.remain$>0&&$t'.deadline$<$l$ **do**     // $t'.remain$ indicates the remaining unallocated workload of $t'$
4.        **If** $t'.remain$ >= $total\_ability$
5.          **For each** node $n'$ in $N_l$ **do**
6.            add < $n'$, $t'$, $l$ > to $t'.assign$
7.            Delete $n'$ from $N_l$
8.          **End For**
9.          $t'.remain$ = $t'.remain$- $total\_ability$
10.         $L$ = $l$ + 1   // all nodes have been assigned to calculate $t'$ at time $l$
11.         Reset $N_l$ // node set at time $l$ + 1
12.        **Else**
13.          $j$ = 1
14.          **While** the $j$th node $n'$ in $N_l$ is not null **do**
15.            **If** $n'.ability$ <= $t'.remain$
16.              add < $n'$, $t'$, $l$ > to $t'. assign$
17.              Delete $n'$ from $N_l$
18.              $t'.remain$ = $t'.remain$-$n'.ability$;
19.            **ElseIf** $t'.remain$ == 0
20.              $t'.cost$ = 0
21.              **Return** $t'.assign$
22.            **Else**
23.              **While** $n'.ability$ >= $t'.remain$&&$n'$ is not null **do**
24.                $j$ = $j$ + 1
25.                $n'$ = the $j$th node in $N_l$
26.              **Endwhile**
27.              $n'$ = the ($j$ − 1)th node in $N_l$
28.              add < $n'$, $t'$, $l$ > to $t'.assign$
29.              Delete $n'$ from $N_l$
30.              $t'.cost$ = $t'.remain$ = 0;
31.            **EndIf**
32.          **EndWhile**
33.        **EndIf**
34.      **EndWhile**
35.      **Return** $t'.assign$

---

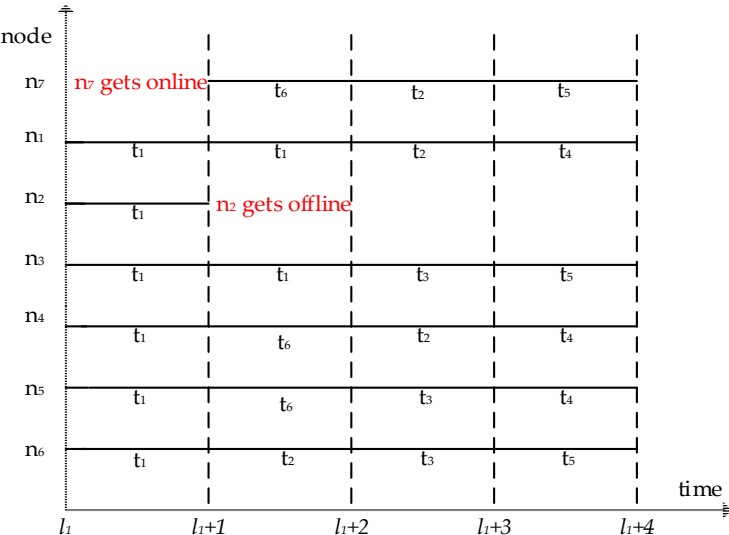

**Figure 4.** The execution processes of the deadline preference dispatch scheduling (DPDS) algorithm.

*4.2. The Improved Dispatch Constrain Scheduling (IDCS) Algorithm*

The DPDS algorithm is a deadline priority allocation method, which can ensure that the most urgent tasks are given the highest priority in dynamic allocation. However, it cannot guarantee the largest number of tasks to be completed. And we find that the task cannot be completed within deadline constraint is still assigned, which causes a waste of computing resources. On this basis, this paper proposes an improved IDCS algorithm. The IDCS algorithm uses a risk prediction model to reduce the waste of computing resources. Before introducing the IDCS algorithm, we introduce the risk prediction model firstly.

4.2.1. The Risk Prediction Model

In this paper, we propose a risk prediction model that can predict the **completion risk** of each task, which is described in Algorithm 3. In VC platforms, VN is constantly updated at every moment. Although, it is impossible to accurately determine the number of updated nodes and their computing power, the range of the number of possible online nodes at each time can be estimated based on historical data. For easy calculation, we assume that the computing power of all predicted online VN is 1 unit, and the probability of the number of possible VN at each time is the same. On this basis, we calculate the **completion risk** of each task by **completion probability**. We will introduce the definition of the **completion probability** below.

**Definition 3 (Completion probability).** *Given a possible world [30] set W and a task t' at time l, the **completion probability** of t' at time l is defined as follows:*

$$\text{Pr}^l(t\prime) = \sum_{w \in W'} \text{Pr}(w), \tag{1}$$

*where w represents a possible world in a possible world set W, W' is a possible world set which is composed of a possible world that can complete t' within the deadline, and Pr(w) represents the **possible probability** of w.*

For example, at time $l_1$, the task $t_6$ is allocated at time $l_1 + 3$, and the deadline of $t_6$ is $l_1 + 4$. Figure 5a shows that the number of possible VN at time $l_1 + 3$ is 3, 4, and 5, and the number of possible VN at time $l_1 + 4$ is 3, 4, and 5. The possible world set $W$ of $t_6$ is shown in Figure 5c. The **completion probability** $\text{Pr}^l_1(t_6)$ of $t_6$ is 66.7%, which is calculated by Equation (1), so the **completion risk** of the task $t_6$ is 33.3%.

---

**Algorithm 3. The Risk Prediction Model Risk(*t*,*W*,*l*)**

---

**Input:** the task *t*, possible world set *W*, the current time *l*
**Output:** the **completion risk** *R* of the task *t*
1.    $l' = t.deadline$
2.    *W* = possible world set from time *l* *to* time *l'*
3.    $Pr^l(t) = 0$
4.    **For each** $w \in W$ **do**
5.       **If** *t* can be completed by *w*
6.          $Pr^l(t) = Pr^l(t) + Pr(w)$
7.       **EndIf**
8.    **EndFor**
9.    $R = 1 - Pr^l(t)$
10.   **Return completion risk** *R* of the task *t*

---

| time | the range of online volunteer nodes |
|------|-------------------------------------|
| $l_1$ | 7-10 |
| $l_1$+1 | 5-8 |
| $l_1$+2 | 4-7 |
| $l_1$+3 | 3-5 |
| $l_1$+4 | 3-5 |

(**a**)

| *T* | $t_i.cost$ | $t_i.deadline$ |
|-----|-----------|----------------|
| $t_1$ | 3 | 4 |
| $t_2$ | 5 | 2 |
| $t_3$ | 6 | 2 |
| $t_4$ | 6 | 4 |
| $t_5$ | 7 | 3 |
| $t_6$ | 8 | 5 |

(**b**)

| possible world | $l_1$+3 | $l_1$+4 | possible probability | completed or not |
|----------------|---------|---------|----------------------|------------------|
| $w_1$ | 3 | 3 | 1/9 | no |
| $w_2$ | 3 | 4 | 1/9 | no |
| $w_3$ | 3 | 5 | 1/9 | yes |
| $w_4$ | 4 | 3 | 1/9 | no |
| $w_5$ | 4 | 4 | 1/9 | yes |
| $w_6$ | 4 | 5 | 1/9 | yes |
| $w_7$ | 5 | 3 | 1/9 | yes |
| $w_8$ | 5 | 4 | 1/9 | yes |
| $w_9$ | 5 | 5 | 1/9 | yes |

(**c**)

**Figure 5.** An example of using a risk prediction model to predict completion risk of the task. (**a**) The range of online volunteer nodes from time $l_1$ to time $l_1 + 4$; (**b**) list of tasks at time $l_1$; (**c**) the possible world set *W* of the task $t_6$.

### 4.2.2. The Description of the IDCS Algorithm

The IDCS algorithm is a dynamic allocation method based on the objective of maximizing the number of tasks completed. To achieve the objective of completing the maximum number of tasks with limited computing resources, the IDCS algorithm chooses the task to assign based on **task priority** and **completion risk** of the task.

Task partitioning is divided into simple and urgent tasks, simple tasks, complex tasks and complex and urgent tasks according to their computational cost and the size of deadline constraints, which correspond to different **task priority**, as shown in Figure 6.

| task classification | task priority |
|---|---|
| simple and urgent task | 1 |
| simple task | 2 |
| complex task | 3 |
| complex and urgent task | 4 |

**Figure 6.** Task classification and corresponding priority.

The specific ways of division are as follows:

If the computational cost of task $t$ is less than the average computational cost of all tasks in the task list, and $t.$ *deadline* is less than the middle time of the whole computing time, it is considered that the task $t$ is simple and urgent, and the **task priority** corresponding to $t$ is 1. If the computational cost of the task $t$ is less than the average computational cost of all tasks in the task list, and $t.$ *deadline* is greater than the middle time of the whole computing time, it is considered that task $t$ is a simple task, and the **task priority** corresponding to $t$ is 2. If the computational cost of task $t$ is greater than the average computational cost of all tasks in the task list and $t.$ *deadline* is greater than the middle time of the whole computing time, it is considered that $t$ is a complex task, and the **task priority** corresponding to $t$ is 3. If the computational cost of task $t$ is greater than the average computational cost of all tasks in the task list, and $t.$ deadline is less than the middle time of the whole computing time, it is considered that task $t$ is complex and urgent, and the **task priority** corresponding to $t$ is 4.

For example, as shown in Figure 5b, the average computational cost of all tasks is 5.8 and the middle time of the whole computing time is 2.5. Therefore, according to the division criteria mentioned above, the task $t_1$ is a simple task and the **task priority** of $t_1$ is 2. The task $t_2$ is a simple and urgent task, and the **task priority** of $t_2$ is 1. The task $t_3$ is a complex and urgent task, and the **task priority** of $t_3$ is 4. The task $t_4$, $t_5$, $t_6$ are complex tasks, and their **task priority** is 3.

In the IDCS algorithm, firstly, the **task priority** and the **completion risk** are calculated as described above. Secondly, the IDCS algorithm deletes tasks that cannot be completed based on **task priority** and **completion risk**. Finally, Algorithm 1 is called to allocate tasks. The IDCS algorithm is described in detail in Algorithm 4.

---

**Algorithm 4. The IDCS Algorithm**

---

**Input:** task set $T_l$, volunteer node set $N_l$, threshold $\theta$, the current time $l$, possible world $W$
**Output:** task assignment set *T.assign*.
1.   Calculate the **task priority** and the **completion risk** of each task in $T_l$
2.   **For** each $t' \in T_l$ **do**
3.     **If** **Risk($t'$,$W$,$l$)**$> \theta$**&&** (**task priority** of $t' = 4$ or **task priority** of $t' = 3$ )
4.       Delete the task $t'$ from $T_l$
5.     **EndIf**
6.   **End For**
7.   **Call taskassign($T_l$, $N_l$, $l$)//** call Algorithm 1
8.   **Return** *T.assign*

---

For example, the list of tasks at time $l_1$ is shown in Figure 2a, and the list of volunteer nodes at time $l_1$ is shown in Figure 2b. List of tasks at time $l_1 + 1$ is shown in Figure 3a, and the list of volunteer nodes at time $l_1 + 1$ is shown in Figure 3b. For the easy calculation, we assume that the nodes and tasks are not updated at other times, and the value of $\theta$ is 0.5. The **completion risk** of $t_3$ is 0.67 according to Algorithm 3, which is assigned at time $l_1 + 3$. According to Algorithm 4, $t_3$ is deleted from the task set $T_l$. According to Algorithm 1, the assignment of the remaining tasks is shown in Figure 7. According to the task allocation of Figure 7, the tasks can be completed within deadline constraint are $t_1$, $t_2$, $t_4$ and

$t_5$. Figure 7 shows that the IDCS algorithm can maximize the number of the completed tasks within deadline constraint.

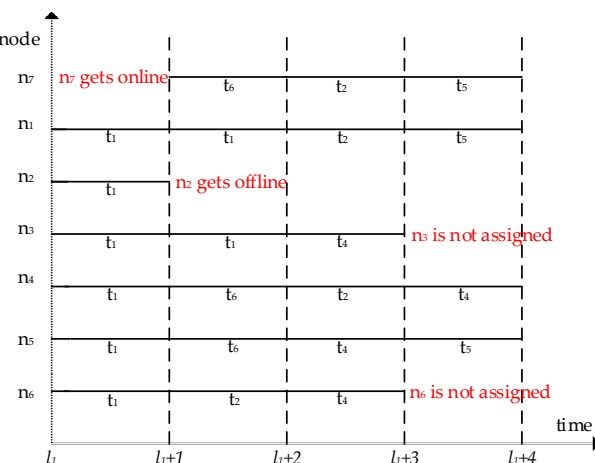

**Figure 7.** The execution processes of the improved dispatch constraint scheduling (IDCS) algorithm.

## 5. Experimental Evaluation

In this section, we first implement the DPDS algorithm and the IDCS algorithm, and use static task set and dynamic task set to compare the performance between the MOC algorithm and our proposed algorithms. The volunteer computing used in the experiment consists of one master node and fifty volunteer nodes. All nodes are configured with Intel Core i7 4790 CPU@3.4GHZ, 8GB DDR3 memory, 1TB hard disk and Windows 10 operating system. To be closer to the real volunteer computing environment and meet the heterogeneity of volunteer computing, 10–20 threads are opened on each host to simulate volunteer nodes, thus, whose number will be between 500 and 1000. Task data fragmentation size is 64 MB, and the parameter $n_j.ability$ is tested by the server sending the applet to the node before assigning tasks.

Specifically, to achieve the heterogeneity of nodes, we use a program to specify different CPU cores for some specific threads. In this way, scheduling delays can be reduced by specifying the CPU core for some specific threads. Thus, the performance of some specific threads will be improved.

### 5.1. Experimental Results and Analysis of the Static Task Sets

In the experiment of the static task set, three common tasks are used: Word frequency statistics, inverted index and distributed Grep. The input files are the data and dump files provided by Wikipedia (the main contents are entries, templates, picture descriptions and basic meta-pages, etc.). We mainly consider the influence of three main parameters as follows:

- The task set scale which is the number of tasks included in the task set $T$.
- The average size of tasks in task set $T$ is measured by the number of task input file fragments.
- The average completion time of tasks in $T$.

We assume that the size of a task set fragment is 64 MB, the threshold $\theta$ is 0.5 and the completion time of each task fragment is 70 s. The task amount of word frequency statistics for a fragment (unit) is 40 s, the task amount of inverted index for a fragment (unit) is 80 s, and the task amount of distributed Grep for a fragment (unit) is 120 s. The average completion time of the tasks in $T$ is 80 s, denoted by $L$. For any task in $T$, $t.\ deadline$ is a random value in the interval $[0.5L, 1.5L]$. Table 2 shows the default values and ranges of the main parameters. Table 2 shows the default values and ranges of the main parameters.

**Table 2.** Experimental default parameters.

| Parameter | Default | Value Range |
|---|---|---|
| average size of tasks(unit) | 4 | 3–9 |
| task set scale | 20 | 10–30 |
| the number of VN | 700 | 500–1000 |

In this paper, the number of the completed tasks is the primary performance index. In addition, this paper also uses the completion rate to measure the performance of the algorithm more comprehensively. The completion rate is defined as follows:

$$\text{completion rate} = \text{number of tasks completed on time/number of tasks in the task set } T, \qquad (2)$$

5.1.1. The Impact of Average Size of Tasks

As shown in Figure 8, we test the impact of the different average size of tasks on the performance of the algorithms. It can be seen that IDCS perform the best among the three algorithms in both the number of the completed tasks and completion rate, and the MOC algorithm is slightly worse than the IDCS algorithm. The DPDS algorithm is much less efficient. This is because the IDCS algorithm divides the task priority, which can ensure that the IDCS algorithm completes the less expensive task first, and discards the risky task. In contrast, the DPDS algorithm can't fully utilize the computing power of the volunteer computing system. Even if it encounters the task that is expensive, the DPDS algorithm will also calculate it, which causes a waste of computing resources. Since the objective of MOC is complete each task before its deadline, it is less efficient than the IDCS algorithm. Moreover, since the computing power of the system is fixed in a certain period of time, the performance of three algorithms decreases as the average size of tasks increases.

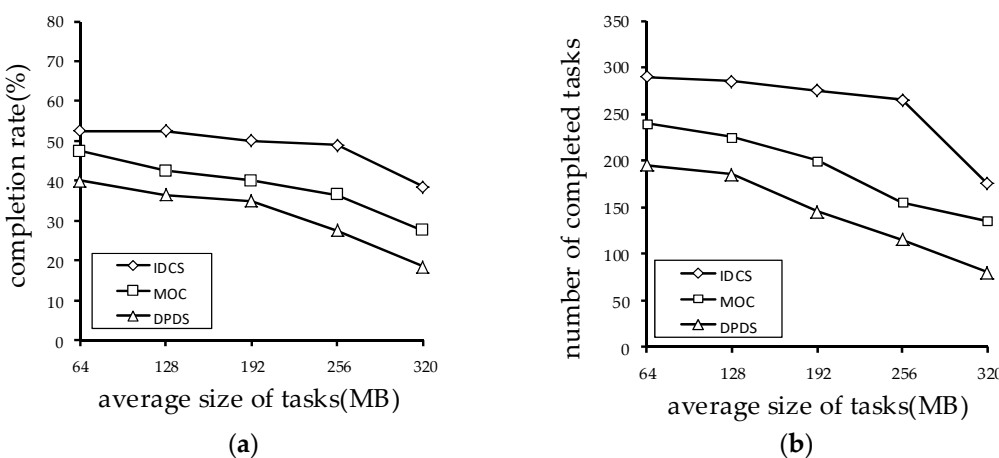

(**a**)  (**b**)

**Figure 8.** The impact of the average size of the tasks on the performance of the algorithms: (**a**) The impact of average size of tasks on task completion rate; (**b**) the impact of the average size of tasks on number of the tasks completed.

5.1.2. The Impact of Task Set Scale

In Figure 9, we analyze the impact of task set scale. It can be seen that with the increase of the task set scale, the number of the completed task of the three algorithms finally tends to be stable. This is because the number of nodes does not increase, but the number of tasks increases. Therefore, the number of the completed tasks by VN is changeless.

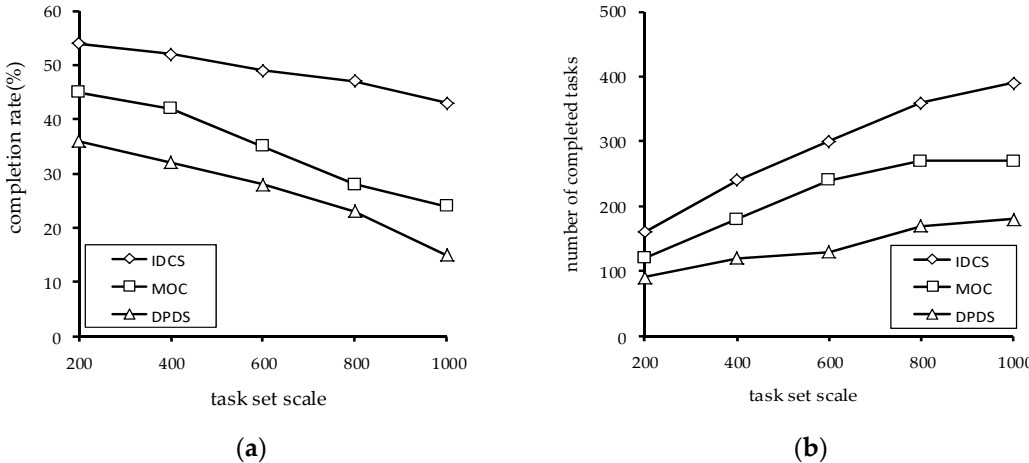

**Figure 9.** The impact of task set scale on the performance of the algorithms: (**a**) The impact of task set scale on task completion rate; (**b**) the impact of the task set scale on the number of tasks completed.

At the same time, it can be seen from Figure 9b that the larger the size of the task set scale is, the more the number of tasks completed by the IDCS algorithm is. This is mainly because IDCS can make full use of computing resources to discards the tasks with high risks, which improves the number of tasks completed.

The task completion rate of the three algorithms decreases with the task set scale increase. This is because the number of tasks has increased, but there has been no relative increase in computing resources. In Figure 9a, it is obvious that the increase of task set scale has the least impact on IDCS algorithm.

### 5.1.3. The Impact of the Number of Volunteer Nodes

Figure 10 shows the impact of the number of volunteer nodes on the performance of the algorithms. It can be seen that the task completion rate increases when the number of volunteer nodes increases, and the number of the completed tasks also increases. This is because the more nodes there are, the more computing power there is. At the same time, the IDCS algorithm is superior to the MOC algorithm and DPBS algorithm in both task completion rate and the number of the completed tasks.

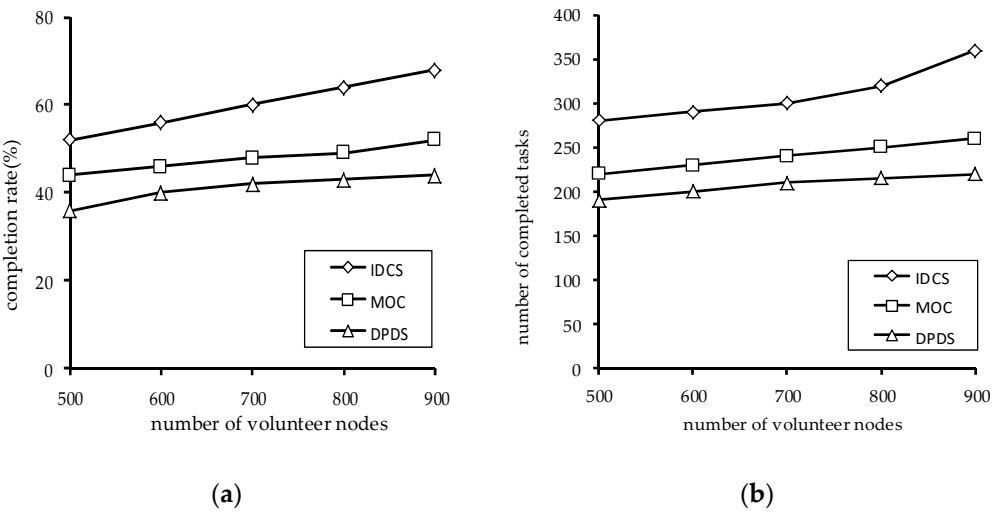

**Figure 10.** The impact of the number of volunteer nodes on the performance of the algorithms: (**a**) The impact of the number of volunteer nodes task completion rate; (**b**) the impact of the number of volunteer nodes on the number of tasks completed.

*5.2. Experimental Results and Analysis of Dynamic Task Sets*

In order to be closer to the real application scenario, this section uses a dynamic set of application tasks. The experiment generated five task sets, and each task set has an average of 100 tasks. The average size of each task is four units. The deadline for each task is set to a random value within 200–400 s after the task arrives. Other parameters settings are the same as the Section 5.1. In the experiment, we assume a task request was submitted to the server every two minutes and the task completion status is tested every 100 min. Figure 11 shows the number of the completed tasks and the task completion rate of the three algorithms.

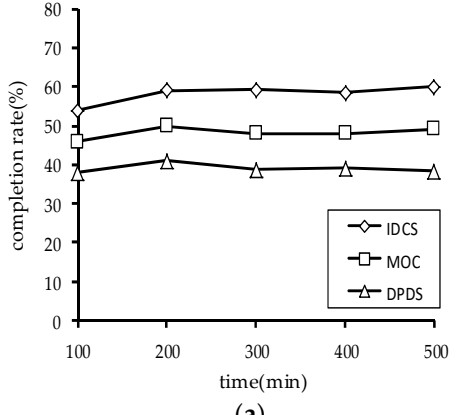
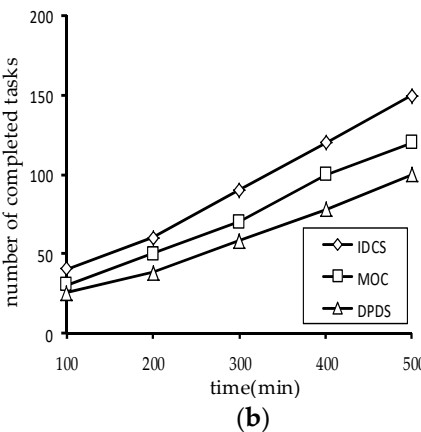

**(a)**　　　　　　　　　　　　　　　　　　　　**(b)**

**Figure 11.** Performance comparisons of the three algorithms on the dynamic task set: (**a**) Comparison of task completion rate; (**b**) comparison of the number of tasks completed.

Figure 11 shows the experimental results. It can be seen that the IDCS algorithm has obvious advantages on dynamic task sets, regardless of the number of the completed tasks or the task completion rate. Through the above experimental results, the validity of the IDCS algorithm proposed in this paper is further proved.

## 6. Conclusions

In this paper, we propose two novel dynamic task scheduling algorithms to solve the task scheduling problem with deadline constraint in VC platforms. One is the DPDS algorithm, and the other is the IDCS algorithm. In addition, by analyzing the characteristics of tasks and volunteer nodes in VC platforms, this paper uses a new risk prediction model in IDCS algorithm, which can predict the completion risk of each task. A lot of experiments based on a real-world dataset demonstrate that our proposed algorithms can solve the dynamic task scheduling problem with deadline constraint. And compared to the existing algorithms, the IDCS algorithm can maximize the number of task completions within the deadline. The task scheduling problem in VC platforms is very important, and further study should be done to improve the scheduling algorithm. In future, we will consider more factors that may affect task scheduling in VC platforms.

**Author Contributions:** L.X. designed and wrote the paper; J.Q. supervised the work; L.X. and W.Z. performed the experiments; S.L. analyzed the data. All authors have read and approved the final manuscript.

**Acknowledgments:** This work was supported by the National Social Science Foundation of China (No. 15BYY028) and Dalian University of Foreign Languages Research Foundation (No. 2015XJQN05).

**Conflicts of Interest:** The authors declare no conflict of interest.

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
