# Peer review of "Dynamic Task Scheduling Algorithm with Deadline Constraint in Heterogeneous Volunteer Computing Platforms"

_futureinternet, doi:10.3390/fi11060121_

Reviewer 1 Report

The authors have addressed the issue reported in the previous review.

1. Some suggestions to rephrase:

(1) Line 121: Suggestion
"However, the objective of MOC is that each task must be completed before its deadline, which is different from our work: to complete as many tasks as possible within their respective deadline constraints"
(2) Line 135: Suggestion
"to complete as many tasks as possible".

2. Some issues with figures

(1) Figure 8: Problem in the x-axis numbering of Figure 8
(2) Figure 9: Problem in the legend of Figure 9(a)
(3) Figure 11: Problem in the x-axis numbering of Figure 11

Overall, I would advise to have the paper proofread by a native English speaker.

Author Response

The authors have very carefully followed all constructive comments, and have tried every possible effort to clean up every issue that was raised by you. We provide below a detailed account on the changes that we have made. For all corrections, we highlight all individual changes in the new manuscript.

Reviewer 2 Report

This manuscript is a revised verions of one that I reviewed for the journal "algorithm" and that resulted on rejected.

The authors have improved the manuscript in relation with my previous comments:

(1) The abstract has been improved and now it is much clearer.

(2) The related work section is now more elaborated.

(3) The contributions are now clearly highlighted in the introduction.

(4) Now, the proposed algorithms are compared with MOC.

But there are still one unclear point:

How the heterogenity of nodes is obtained by starting a different number of threads? Please justify more deeply this design choice.

Author Response

The authors have very carefully followed all constructive comments, and have tried every possible effort to clean up every issue that was raised by you. We provide below a detailed account on the changes that we have made.

For all corrections, we highlight all individual changes in the new manuscript.

This manuscript is a resubmission of an earlier submission.

The following is a list of the peer review reports and author responses from that submission.

Round  1

Reviewer 1 Report

The authors propose two algorithms for the problem of dynamic task allocation with deadlines in heterogeneous volunteer networks.

Important weakness have been detected.

The contribution of the authors is not clear. Are the two algorithms new contributions? Are they based in other previous strategies/algorithms applied in other domains different to the VC? The authors should clarify, contributions and the comparison with previous solutions from both the domain of VC or similar algorithms from other domains. 

The related work section needs improvements. The authors just explain the previous work in a isolated way, and they are not compared with their proposal and other previous. Apart from a rewriting of the section, a table which the comparison of features could be help readers.

The abstract should be improved. We recommend to the authors to check strategies to write suitable abstracts and rewrite it.

 The english of the manuscript should be improved. We recommend that the text will be checked by an english-native speaker.

The result section is not enough, since the authors only compares the results of their two proposals. Consequently, they are only proving that one of their algorithms is better than the other. They must compare the solutions with a third solution of some of the previous studies.

The solution to generate heterogeneous nodes is not realistic. If the authors only have homogenic nodes, I recommend them to generate one Virtual Machine in each node with different memory and number of vcpu. From my point of view, this is more realistic than including sleeps in the tasks. 

Other minor comments:

Table 1 does not include the parameter nj.ability

Parameter nj.ability is writting wrong several times along the manuscript.

How this parameter would be calculated in a real scenarios, where only the hardware features of the nodes are provided? Additionally, the other workloads of the VC nodes influence in the performance of the node. The authors should elaborate how this parameter could be determined in real systems.

Reviewer 2 Report

The paper presents two algorithms for dynamic task scheduling over volunteer

computing resources, when tasks have deadline constraints. The algorithms are: 

i) Deadline Preference Dispatch Scheduling (DPDS)

ii) Improved Dispatch Constrain Scheduling (IDCS)

DPDS schedules the task with the smallest deadline to the computing resource that has the highest computing power, while IDCS prioritizes completion of tasks within the defined deadline. The paper is properly structured, but the English should be improved, since some sentences are difficult to understand (see below).

The main weakness of the paper is the lack of proper results comparison. Indeed, the experimental evaluation solely compares the two algorithms -- DPDS and IDCS -- with each other. So, how does DPDS and IDCS compare with other algorithms for task scheduling in volunteer computing platforms? 

Moreover, the experimental evaluation does not tackle the scalability of the proposed scheduling algorithms, since the number of volunteer resources is limited to one master node and four volunteer computing nodes. Volunteer computing is known for having large number of computing nodes, in the ranges of thousands, if not more. Therefore, four volunteer nodes is too limited for a proper evaluation, even if the algorithms are targeting solely local volunteer resources (e.g., resources of a local network).

Regarding DPDS, the following two sentences are hard to interpret:

"Secondly the DPDS algorithm adopts a matching function to select the most suitable volunteer node to assign task, which can waste the least computing power of VN. Consequently, the DPDS algorithm can ensure that the task with minimum deadline constraint is completed first, and waste the least computing power of VN." 

What do you exactly mean by "waste the least computing power" and why? Is it because that by selecting the most powerful available computing resource, the task will be completed within the shortest delay, and therefore the computing resource will be freed in the least possible amount of time? If so, both sentences should be made clearer, avoiding the word "waste" (for instance, using "minimize the time required for finishing the task, and thus freeing the volunteer resource for other tasks")

Another confusing sentence is in page 11: 

- "With the increase of time, the number of nodes and tasks submitted will increase, and the number of tasks completed will gradually increase with the increase of time." 

What does exactly mean "increase of time"? Lengthier tasks? A larger time interval? Please, clarify.

Some observations:

==================

- The keyword "Possible world" is a bit awkward

- Some sentences need to be rewritten to ensure proper clarity. 

- Contrary to the other plots of the paper, plot of Figure 11 b) uses the same square symbol for both DPDS and IDCS.